# Real-Time Inference for a Gamma Process Model of Neural Spiking

[1]**David Carlson,** [2]**Vinayak Rao,** [2]**Joshua Vogelstein,** [1]**Lawrence Carin**
[1]Electrical and Computer Engineering Department, Duke University
[2]Statistics Department, Duke University
{dec18,lcarin}@duke.edu, {var11,jovo}@stat.duke.edu

## Abstract

With simultaneous measurements from ever increasing populations of neurons, there is a growing need for sophisticated tools to recover signals from individual neurons. In electrophysiology experiments, this classically proceeds in a two-step process: (i) threshold the waveforms to *detect* putative spikes and (ii) *cluster* the waveforms into single units (neurons). We extend previous Bayesian nonparametric models of neural spiking to *jointly* detect and cluster neurons using a Gamma process model. Importantly, we develop an online approximate inference scheme enabling real-time analysis, with performance exceeding the previous state-of-the-art. Via exploratory data analysis—using data with partial ground truth as well as two novel data sets—we find several features of our model collectively contribute to our improved performance including: (i) accounting for colored noise, (ii) detecting overlapping spikes, (iii) tracking waveform dynamics, and (iv) using multiple channels. We hope to enable novel experiments simultaneously measuring many thousands of neurons and possibly adapting stimuli dynamically to probe ever deeper into the mysteries of the brain.

## 1 Introduction

The recent heightened interest in understanding the brain calls for the development of technologies that will advance our understanding of neuroscience. Crucial for this endeavor is the advancement of our ability to understand the *dynamics* of the brain, via the measurement of large populations of neural activity at the single neuron level. Such reverse engineering efforts benefit from real-time decoding of neural activity, to facilitate effectively adapting the probing stimuli. Regardless of the experimental apparati used (e.g., electrodes or calcium imaging), real-time decoding of individual neuron responses requires identifying and labeling individual spikes from recordings from large populations. In other words, real-time decoding requires real-time spike sorting.

Automatic spike sorting methods are continually evolving to deal with more sophisticated experiments. Most recently, several methods have been proposed to (i) learn the number of separable neurons on each electrode or "multi-trode" [1, 2], or (ii) operate online to resolve overlapping spikes from multiple neurons [3]. To our knowledge, no method to date is able to simultaneously address both of these challenges.

We develop a nonparametric Bayesian continuous-time generative model of population activity. Our model explains the continuous output of each neuron by a latent marked Poisson process, with the "marks" characterizing the shape of each spike. Previous efforts to address overlapping spiking often assume a fixed kernel for each waveform, but joint intracellular and extracellular recording clearly indicate that this assumption is false (see Figure 3c). Thus, we assume that the statistics of the marks are time-varying. We use the framework of completely random measures to infer how many of a potentially infinite number of neurons (or single units) are responsible for the observed data, simultaneously characterizing spike times and waveforms of these neurons

We describe an intuitive discrete-time approximation to the above infinite-dimensional continuous-time stochastic process, then develop an online variational Bayesian inference algorithm for this model. Via numerical simulations, we demonstrate that our inference procedure improves

over the previous state-of-the-art, even though we allow the other methods to use the entire dataset for training, whereas we learn online. Moreover, we demonstrate that we can effectively track the time-varying changes in waveform, and detect overlapping spikes. Indeed, it seems that the false positive detections from our approach have indistinguishable first order statistics from the true positives, suggesting that second-order methods may be required to reduce the false positive rate (i.e., template methods may be inadequate). Our work therefore suggests that further improvements in real-time decoding of activity may be most effective if directed at simultaneous real-time spike sorting and decoding. To facilitate such developments and support reproducible research, all code and data associated with this work is provided in the Supplementary Materials.

## 2   Model

Our data is a time-series of multielectrode recordings $\mathbf{X} \equiv (\mathbf{x}_1, \cdots, \mathbf{x}_T)$, and consists of $T$ recordings from $M$ channels. As in usual measurement systems, the recording times lie on regular grid, with interval length $\Delta$, and $\mathbf{x}_t \in \mathbb{R}^M$ for all $t$. Underlying these observations is a continuous-time electrical signal driven by an unknown number of neurons. Each neuron generates a continuous-time voltage trace, and the outputs of all neurons are superimposed and discretely sampled to produce the recordings $\mathbf{X}$. At a high level, in §2.1 we model the continuous-time output of each neuron as a series of idealized Poisson events smoothed with appropriate kernels, while §2.2 uses the Gamma process to develop a nonparametric prior for an entire population. §2.3 then describes a discrete-time approximation based on the Bernoulli approximation to the Poisson process. For conceptual clarity, we restrict ourselves to single channel recordings until §2.4, where we describe the complete model for multichannel data.

### 2.1   Modeling the continuous-time output of a single neuron

There is a rich literature characterizing the spiking activity of a single neuron [4] accounting in detail for factors like non-stationarity, refractoriness and spike waveform. We however make a number of simplifying assumptions (some of which we later relax). First, we model the spiking activity of each neuron are stationary and memoryless, so that its set of spike times are distributed as a homogeneous Poisson process (PP). We model the neurons themselves are heterogeneous, with the $i^{th}$ neuron having an (unknown) firing rate $\lambda_i$. Call the ordered set of spike times of the $i^{th}$ neuron $\mathcal{T}_i = (\tau_{i1}, \tau_{i2}, \ldots)$; then the time between successive elements of $\mathcal{T}_i$ is exponentially distributed with mean $1/\lambda_i$. We write this as $\mathcal{T}_i \sim \mathrm{PP}(\lambda_i)$.

The actual electrical output of a neuron is not binary; instead each spiking event is a smooth perturbation in voltage about a resting state. This perturbation forms the shape of the spike, with the spike shapes varying across neurons as well as across different spikes of the same neuron. However, each neuron has its own characteristic distribution over shapes, and we let $\boldsymbol{\theta}_i^* \in \Theta$ parametrize this distribution for neuron $i$. Whenever this neuron emits a spike, a new shape is drawn independently from the corresponding distribution. This waveform is then offset to the time of the spike, and contributes to the voltage trace associated with that spike.

The complete recording from the neuron is the superposition of all these spike waveforms plus noise. Rather than treating the noise as white as is common in the literature [5], we allow it to exhibit temporal correlation, recognizing that the 'noise' is in actual fact background neural activity. We model it as a realization of a Gaussian process (GP) [6], with the covariance kernel $\mathcal{K}$ of the GP determining the temporal structure. We use an exponential kernel, modeling the noise as Markov.

We model each spike shape as weighted superpositions of a dictionary of $K$ basis functions $\mathbf{d}(t) \equiv (\mathsf{d}_1(t), \cdots, \mathsf{d}_K(t))^\mathsf{T}$. The dictionary elements are shared across all neurons, and each is a real-valued function of time, i.e., $\mathsf{d}_k \in L_2$. Each spike time $\tau_{ij}$ is associated with a random $K$-dimensional weight vector $\mathbf{y}_{ij}^* \equiv (y_{ij1}^*, \ldots y_{ijK}^*)^\mathsf{T}$, and the shape of this spike at time $t$ is given by the weighted sum $\sum_{k=1}^K y_{ijk}^* \mathsf{d}_k(t - \tau_{ij})$. We assume $\mathbf{y}_{ij}^* \sim \mathsf{N}_K(\boldsymbol{\mu}_i^*, \Sigma_i^*)$, indicating a $K$-dimensional Gaussian distribution with mean and covariance given by $(\boldsymbol{\mu}_i^*, \Sigma_i^*)$; we let $\theta_i^* \equiv (\boldsymbol{\mu}_i^*, \Sigma_i^*)$. Then, at any time $t$, the output of neuron $i$ is $x_i(t) = \sum_{j=1}^{|\mathcal{T}_i|} \sum_{k=1}^K y_{ijk}^* \mathsf{d}_k(t - \tau_{ij})$.

The total signal received by any electrode is the superposition of the outputs of all neurons. Assume for the moment there are $N$ neurons, and define $\mathcal{T} \equiv \cup_{i \in [N]} \mathcal{T}_i$ as the (ordered) union of the spike times of all neurons. Let $\tau_l \in \mathcal{T}$ indicate the time of the $l^{th}$ overall spike, whereas $\tau_{ij} \in \mathcal{T}_i$ is the time of the $j^{th}$ spike of neuron $i$. This defines a pair of mappings: $\nu : [|\mathcal{T}|] \to [N]$, and $p : [|\mathcal{T}|] \to \mathcal{T}_{\nu_i}$, with $\tau_l = \tau_{\nu_l p_l}$. In words, $\nu_l \in N$ is the neuron to which the $l^{th}$ element of $\mathcal{T}$ belongs, while $p_l$ indexes this spike in the spike train $\mathcal{T}_{\nu_l}$. Let $\boldsymbol{\theta}_l \equiv (\boldsymbol{\mu}_l, \Sigma_l)$ be the neuron parameter associated with spike $l$, so that $\boldsymbol{\theta}_l = \boldsymbol{\theta}_{\nu_l}^*$. Finally, define $\mathbf{y}_l \equiv (y_{l1}, \ldots, y_{lK})^\mathsf{T} \equiv \mathbf{y}_{\nu_j p_j}^*$ as the weight

vector of spike $\tau_l$. Then, we have that

$$x(t) = \sum_{i \in [N]} x_i(t) = \sum_{l \in |\mathcal{T}|} \sum_{k \in [K]} y_{lk} \mathsf{d}_k(t - \tau_l), \qquad \text{where } \mathbf{y}_l \sim \mathsf{N}_K(\boldsymbol{\mu}_l, \Sigma_l). \qquad (1)$$

From the superposition property of the Poisson process [7], the overall spiking activity $\mathcal{T}$ is Poisson with rate $\Lambda = \sum_{i \in [N]} \lambda_i$. Each event $\tau_l \in \mathcal{T}$ has a pair of labels, its neuron parameter $\boldsymbol{\theta}_l \equiv (\boldsymbol{\mu}_l, \Sigma_l)$, and $\mathbf{y}_l$, the weight-vector characterizing the spike shape. We view these weight-vectors as the "marks" of a marked Poisson process $\mathcal{T}$. From the properties of the Poisson process, we have that the marks $\boldsymbol{\theta}_l$ are drawn i.i.d. from a probability measure $\mathsf{G}(\mathrm{d}\boldsymbol{\theta}) = 1/\Lambda \sum_{i \in [N]} \lambda_i \delta_{\boldsymbol{\theta}_i^*}$.

With probability one, the neurons have distinct parameters, so that the mark $\boldsymbol{\theta}_l$ identifies the neuron which produced spike $l$: $\mathsf{G}(\boldsymbol{\theta}_l = \boldsymbol{\theta}_i^*) = \mathsf{P}(\nu_l = i) = \lambda_i/\Lambda$. Given $\boldsymbol{\theta}_l$, $\mathbf{y}_l$ is distributed as in Eq. (1). The output waveform $x(t)$ is then a linear functional of this marked Poisson process.

## 2.2 A nonparametric model of population activity

In practice, the number of neurons driving the recorded activity is unknown. We do not wish to bound this number *a priori*, moreover we expect this number to increase as we record over longer intervals. This suggests a nonparametric Bayesian approach: allow the *total* number of underlying neurons to be infinite. Over any finite interval, only a finite subset of these will be *active*, and typically, these dominate spiking activity over any interval. This elegant and flexible modeling approach allows the data to suggest how many neurons are active, and has already proved successful in neuroscience applications [8]. We use the framework of *completely random measures* (CRMs) [9] to model our data. CRMs have been well studied in the Bayesian nonparametrics community, and there is a wealth of literature on theoretical properties, as well as posterior computation; see e.g. [10, 11, 12]. Recalling that each neuron is characterized by a pair of parameters $(\lambda_i, \boldsymbol{\theta}_i^*)$, we map the infinite collection of pairs $\{(\lambda_i, \boldsymbol{\theta}_i^*)\}$ to an random measure $\Lambda(\cdot)$ on $\Theta$: $\Lambda(\mathrm{d}\boldsymbol{\theta}) = \sum_{i=1}^{\infty} \lambda_i \delta_{\boldsymbol{\theta}_i^*}$.

For a CRM, the distribution over measures is induced by distributions over the infinite sequence of weights, and the infinite sequence of their locations. The weights $\lambda_i$ are the jumps of a Lévy process [13], and their distribution is characterized by a Lévy measure $\rho(\lambda)$. The locations $\boldsymbol{\theta}_i^*$ are drawn i.i.d. from a base probability measure $\mathsf{H}(\boldsymbol{\theta}^*)$. As is typical, we assume these to be independent.

We set the Lévy measure $\rho(\lambda) = \alpha \lambda^{-1} \exp(-\lambda)$, resulting in a CRM called the Gamma process (ΓP) [14]. The Gamma process has the convenient property that the total rate $\Lambda \equiv \Lambda(\Theta) = \sum_{i=1}^{\infty} \lambda_i$ is Gamma distributed (and thus conjugate to the Poisson process prior on $\mathcal{T}$). The Gamma process is also closely connected with the Dirichlet process [15], which will prove useful later on. To complete the specification on the Gamma process, we set $\mathsf{H}_\phi(\boldsymbol{\theta}^*)$ to the conjugate normal-Wishart distribution with hyperparameters $\phi$.

It is easy to directly specify the resulting continuous-time model, we provide the equations in the Supplementary Material. However it is more convenient to represent the model using the marked Poisson process of Eq. (1). There, the overall process $\mathcal{T}$ is a rate $\Lambda$ Poisson process, and under a Gamma process prior, $\Lambda$ is Gamma$(\alpha, 1)$ distributed [15]. The labels $\boldsymbol{\theta}_i$ assigning events to neurons are drawn i.i.d. from a normalized Gamma process: $\mathsf{G}(\mathrm{d}\boldsymbol{\theta}) = (1/\Lambda) \sum_{l=1}^{\infty} \lambda_l$.

$\mathsf{G}(\mathrm{d}\boldsymbol{\theta})$ is a random probability measure (RPM) called a *normalized random measure* [10]. Crucially, a normalized Gamma process is the Dirichlet process (DP) [15], so that the spike parameters $\boldsymbol{\theta}$ are i.i.d. draws with a DP-distributed RPM. For spike $l$, the shape vector is drawn from a normal with parameters $(\boldsymbol{\mu}_l, \Sigma_l)$: these are thus draws from a DP mixture (DPM) of Gaussians [16].

We can exploit the connection with the DP to integrate out the infinite-dimensional measure $\mathsf{G}(\cdot)$ (and thus $\Lambda(\cdot)$), and assign spikes to neurons via the so-called Chinese restaurant process (CRP) [17]. Under this scheme, the $l^{th}$ spike is assigned the same parameter as an earlier spike with probability proportional to the number of earlier spikes having that parameter. It is assigned a new parameter (and thus, a new neuron is observed) with probability proportional to $\alpha$. Letting $C_t$ be the number of neurons observed until time $t$, and $\mathcal{T}_i^t = \mathcal{T}_i \cap [0, t)$ be the times of spikes produced by neuron $i$ before time $t$, we then have for spike $l$ at time $t = \tau_l$:

$$\boldsymbol{\theta}_l = \boldsymbol{\theta}_{\nu_l}^*, \text{ where } P(\nu_l = i) \propto \begin{cases} |\mathcal{T}_i^t| & i \in [C_t], \\ \alpha & i = C_t + 1, \end{cases} \qquad (2)$$

This marginalization property of the DP allows us to integrate out the infinite-dimensional rate vector $\Lambda(\cdot)$, and sequentially assign spikes to neurons based on the assignments of earlier spikes. This requires one last property: for the Gamma process, the RPM $\mathsf{G}(\cdot)$ is independent of the total mass $\Lambda$. Consequently, the clustering of spikes (determined by $\mathsf{G}(\cdot)$) is independent of the rate $\Lambda$ at which they are produced. We then have the following model:

$$\mathcal{T} \sim \mathsf{PP}(\Lambda), \qquad\qquad\qquad \text{where } \Lambda \sim \mathsf{\Gamma P}(\alpha, 1), \qquad\qquad (3a)$$

$$\mathbf{y}_l \sim \mathsf{N}_K(\boldsymbol{\mu}_l, \Sigma_l), \qquad\qquad \text{where } (\boldsymbol{\mu}_l, \Sigma_l) \sim \mathsf{CRP}(\alpha, \mathsf{H}_\phi(\cdot)), \quad l \in [|\mathcal{T}|], \quad (3b)$$

$$x(t) = \sum_{l \in |\mathcal{T}|} \sum_{k \in [K]} y_{lk} \mathsf{d}_k(t - \tau_l) + \varepsilon_t \qquad \text{where } \varepsilon \sim \mathsf{GP}(0, \mathcal{K}). \qquad\qquad (3c)$$

## 2.3 A discrete-time approximation

The previous subsections modeled the continuous-time voltage output of a neural population. Our data on the other hand consists of recordings at a discrete set of times. While it is possible to make inferences about the continuous-time process underlying these discrete recordings, in this paper, we restrict ourselves to the discrete case. The marked Poisson process characterization of Eq. 3 leads to a simple discrete-time approximation of our model.

Recall first the Bernoulli approximation to the Poisson process: a sample from a Poisson process with rate $\Lambda$ can be approximated by discretizing time at a granularity $\Delta$, and assigning each bin an event independently with probability $\Lambda\Delta$ (the accuracy of the approximation increasing as $\Delta$ tends to 0). To approximate the *marked* Poisson process $\mathcal{T}$, all that is additionally required is to assign marks $\boldsymbol{\theta}_i$ and $\mathbf{y}_i$ to each event in the Bernoulli approximation. Following Eqs. (3b) and (3c), the $\boldsymbol{\theta}_l$'s are distributed according to a Chinese restaurant process, while each $\mathbf{y}_l$ is drawn from a normal distribution parametrized by the corresponding $\boldsymbol{\theta}_l$. We discretize the elements of dictionary as well, yielding discrete dictionary elements $\widetilde{\mathbf{d}}_{k,:} = (\widetilde{d}_{k,1}, \ldots, \widetilde{d}_{k,L})^{\mathsf{T}}$. These form the rows of a $K \times L$ matrix $\widetilde{\mathbf{D}}$ (we call its columns $\widetilde{\mathbf{d}}_{:,h}$). The shape of the $j^{th}$ spike is now a vector of length $L$, and for a weight vector $\mathbf{y}$, is given by $\widetilde{\mathbf{D}}\mathbf{y}$.

We can simplify notation a little for the discrete-time model. Let $t$ index time-bins (so that for an observation interval of length $T$, $t \in [T/\Delta]$). We use tildes for variables indexed by bin-position. Thus, $\widetilde{\nu}_t$ and $\widetilde{\theta}_t$ are the neuron and neuron parameter associated with time bin $t$, and $\widetilde{\mathbf{y}}_t$ is its weight-vector. Let the binary variable $\widetilde{z}_t$ indicate whether or not a spike is present in time bin $t$ (recall that $\widetilde{z}_t \sim \mathsf{Bernoulli}(\Lambda\Delta)$). If there is no spike associated with bin $t$, then we ignore the marks $\widetilde{\mu}$ and $\widetilde{\mathbf{y}}$. Thus the output at time $t$, $x_t$ is given by $x_t = \sum_{h=1}^{L} \widetilde{z}_{t-h} \mathbf{d}_{:,h}^{\mathsf{T}} \widetilde{\mathbf{y}}_{t-h-1} + \varepsilon_t$. Note that the noise $\varepsilon_t$ is now a discrete-time Markov Gaussian process. Let $a$ and $r_t$ be the decay and innovation of the resulting autoregressive (AR) process, so that $\varepsilon_{t+1} = a\varepsilon_t + r_t$.

## 2.4 Correlations in time and across electrodes

So far, for simplicity, we restricted our model to recordings from a single channel. We now describe the full model we use in experiments with multichannel recordings. We let every spike affect the recordings at all channels, with the spike shape varying across channels. For spike $l$ in channel $m$, call the weight-vector $\mathbf{y}_l^m$. All these vectors must be correlated as they correspond to the same spike; we do this simply by concatenating the set of vectors into a single $MK$-element vector $\mathbf{y}_l = (\mathbf{y}_l^1; \cdots ; \mathbf{y}_l^M)$, and modeling this as a multivariate normal. In principle, one might expect the associated covariance matrix to possess a block structure (corresponding to the subvector associated with each channel); however, rather than building this into the model, we allow the data to inform us about any such structure.

We also relax the requirement that the parameters $\boldsymbol{\theta}^*$ of each neuron remain constant, and instead allow $\boldsymbol{\mu}^*$, the mean of the weight-vector distribution, to evolve with time (we keep the covariance parameter $\boldsymbol{\Sigma}_i^*$ fixed, however). Such flexibility can capture effects like changing cell characteristics or moving electrodes. Like the noise term, we model the time-evolution of this quantity as a realization of a Markov Gaussian process; again, in discrete-time, this corresponds to a simple first-order AR process. With $\boldsymbol{B} \in \mathbb{R}^{K \times K}$ the transition matrix, and $\mathbf{r}_t \in \mathbb{R}^K$, independent Gaussian innovations, we have $\boldsymbol{\mu}_{t+1}^* = \mathbf{B}\boldsymbol{\mu}_t^* + \mathbf{r}_t$. Where we previously had a DP mixture of Gaussians, we now have a DP mixture of GPs. Each neuron is now associated with a vector-valued function $\boldsymbol{\theta}^*(\cdot)$, rather than a constant. When a spike at time $\tau_l$ is assigned to neuron $i$, it is assigned a weight-vector $\mathbf{y}_l$ drawn from a Gaussian with mean $\boldsymbol{\mu}_i^*(\tau_l)$. Algorithm 1 in the Supplementary Material summarizes the full generative mechanism for the full discrete-time model.

# 3 Inference

There exists a vast literature on computational approaches to posterior inference for Bayesian non-parametric models, especially so for models based on the DP. Traditional approaches are sampling-based, typically involving Markov chain Monte Carlo techniques (see eg. [18, 19]), and recently there has also been work on constructing deterministic approximations to the intractable posterior (eg. [20, 21]). Our problem is complicated by two additional factors. The first is the convolutional nature of our observation process, where at each time, we observe a function of the previous obser-

vations drawn from the DPMM. This is in contrast to the usual situation where one directly observes the DPMM outputs themselves. The second complication is a computational requirement: typical inference schemes are batch methods that are slow and computationally expensive. Our ultimate goal, on the other hand, is to perform inference in real time, making these approaches unsuitable. Instead, we develop an online algorithm for posterior inference. Our algorithm is inspired by the sequential update and greedy search (SUGS) algorithm of [22], though that work was concerned with the usual case of i.i.d. observations from a DPMM. We generalize SUGS to our observation process, also accounting for the time-evolution of the cluster parameters and correlated noise.

Below, we describe a single iteration of our algorithm for the case a single electrode; generalizing to the multielectrode case is straightforward. At each time $t$, our algorithm maintains the set of times of the spikes it has inferred from the observations so far. It also maintains the identities of the neurons that it assigned each of these spikes to, as well as the weight vectors determining the shapes of the associated spike waveforms. We indicate these point estimates with the hat operator, so, for example $\widehat{\mathcal{T}}_i^t$ is the set of estimated spike times before time $t$ assigned to neuron $i$. In addition to these point estimates, the algorithm also keeps a set of posterior distributions $q_{it}(\theta_i^*)$ where $i$ spans over the set of neurons seen so far (i.e. $i \in [\widehat{C}_t]$). For each $i$, $q_{it}(\theta_i^*)$ approximates the distribution over the parameters $\theta_i^* \equiv (\mu_i^*, \Sigma_i^*)$ of neuron $i$ given the observations until time $t$.

Having identified the time and shape of spikes from earlier times, we can calculate their contribution to the recordings $\mathbf{x}_t^L \equiv (x_t, \cdots, x_{t+L-1})^\mathsf{T}$. Recalling that the basis functions $\mathbf{D}$, and thus all spike waveforms, span $L$ time bins, the residual at time $t + t_1$ is then given by $\delta x_{t+t_1} = x_t - \sum_{h \in [L-t_1]} \widehat{z}_{t-h} \mathbf{D} \widehat{\mathbf{y}}_{t-h}$ (at time $t$, for $t_1 > 0$, we define $\widehat{z}_{t+t_1} = 0$). We treat the residual $\delta\mathbf{x}_t = (\delta x_t, \cdots, \delta x_{t+L})^\mathsf{T}$ as an observation from a DP mixture model, and use this to make hard decisions about whether or not this was produced by an underlying spike, what neuron that spike belongs to (one of the earlier neurons or a new neuron), and what the shape of the associated spike waveform is. The latter is used to calculate $q_{i,t+1}(\theta_i^*)$, the new distribution over neuron parameters at time $t + 1$. Our algorithm proceeds recursively in this manner.

For the first step we use Bayes' rule to decide whether there is a spike underlying the residual:

$$\mathsf{P}(\widetilde{z}_t = 1 | \delta\mathbf{x}_t) \propto \sum_{i \in \widehat{C}_t + 1} \mathsf{P}(\delta\mathbf{x}_t, \nu_t = i | \widetilde{z}_t = 1) \mathsf{P}(\widetilde{z}_t = 1) \tag{4}$$

Here, $\mathsf{P}(\delta\mathbf{x}_t | \nu_t = i, \widetilde{z}_t = 1) = \int_\Theta \mathsf{P}(\delta\mathbf{x}_t | \theta_t) q_{it}(\theta_t) \mathrm{d}\theta_t$, while $\mathsf{P}(\nu_t = i | \widetilde{z}_t = 1)$ follows from the CRP update rule (equation (2)). $\mathsf{P}(\delta\mathbf{x}_t | \theta_t)$ is just the normal distribution, while we restrict $q_{it}(\cdot)$ be the family of normal-Wishart distribution. We can then evaluate the integral, and then summation (4) to approximate $\mathsf{P}(\widetilde{z}_t = 1 | \delta\mathbf{x}_t)$. If this exceeds a threshold of $0.5$ we decide that there is a spike present at time $t$, otherwise, we set $\widetilde{z}_t = 0$. Observe that making this decision involves marginalizing over all possible cluster assignments $\nu_t$, and all values of the weight vector $\mathbf{y}_t$. On the other hand, having made this decision, we collapse these posterior distributions to point estimates $\widehat{\nu}_t$ and $\widehat{\mathbf{y}}_t$ equal to their MAP values.

In the event of a spike ($\widehat{z}_t = 1$), we use these point estimates to update the posterior distribution over parameters of cluster $\widehat{\nu}_t$, to obtain $q_{i,t+1}(\cdot)$ from $q_{i,t}(\cdot)$; this is straightforward because of conjugacy. We follow this up with an additional update step for the distributions of the means of *all* clusters: this is to account for the AR evolution of the cluster means. We use a variational update to keep $q_{i,t+1}(\cdot)$ in the normal-Wishart distribution. Finally we take a stochastic gradient step to update any hyperparameters we wish to learn. We provide all details in the Supplementary material.

## 4   Experiments

**Data:**   In the following, we refer to our algorithm as OPASS[1]. We used two different datasets to demonstrate the efficacy of OPASS. First, the ever popular, publicly available HC1 dataset as described in [23]. We used the dataset d533101 that consisted of an extracellular tetrode and a single intracellular electrode. The recording was made simultaneously on all electrodes and was set up such that the cell with the intracellular electrode was also recorded on the extracellular array implanted in the hippocampus of an anesthetized rat. The intracellular recording is relatively noiseless and gives nearly certain firing times of the intracellular neuron. The extracellular recording contains the spike waveforms from the intracellular neuron as well as an unknown number of additional neurons. The data is a 4-minute recording at a 10 kHz sampling rate.

The second dataset comes from novel NeuroNexus devices implanted in the rat motor cortex. The data was recorded at 32.5 kHz in freely-moving rats. The first device we consider is a set of

3 channels of data (Fig. 7a). The neighboring electrode sites in these devices have 30 $\mu$m between electrode edges and 60 $\mu$m between electrode centers. These devices are close enough that a locally-firing neuron could appear on multiple electrode sites [2], so neighboring channels warrant joint processing. The second device has 8-channels (see Fig. 10a), but is otherwise similar to the first. We used a 15-minute segment of this data for our experiments.

For both datasets, we preprocessed with a high-pass filter at 800 Hz using a fourth order Butterworth filter before we analyzed the time series. To define $\mathbf{D}$, we used the first five principle components of all spikes detected with a threshold (three times the standard deviation of the noise above the mean) in the first five seconds. The noise standard deviation was estimated both over the first five seconds of the recording as well as the entire recording, and the estimate was nearly identical. Our results were also robust to minor variations in the choice of the number of principal components. The autoregressive parameters were estimated by using lag-1 autocorrelation on the same set of data. For the multichannel algorithms we estimate the covariance between channels and normalize by our noise variance estimate.

Each algorithm gives a clustering of the detected spikes. In this dataset, we only have a partial ground truth, so we can only verify accuracy for the neuron with the intracellular (IC) recording. We define a detected spike to be an IC spike if the IC recording has a spike within 0.5 milliseconds (ms) of the detected spike in the extracellular recording. We define the cluster with the greatest number of intracellular spikes as a "IC cluster". We refer to these data as "partial ground truth data", because we know the ground truth spike times for one of the neurons, but not all the others.

**Algorithm Comparisons** We compare a number of variants of OPASS, as well as several previously proposed methods, as described below. The vanilla version of OPASS operates on a single channel with colored noise. When using multiple channels, we append an "M" to obtain MOPASS. When we model the mean of the waveforms as an auto-regressive process, we "post-pend" to obtain OPASSR. We compare these variants of OPASS to Gaussian mixture models and k-means [5] with N components (GMM-N and K-N, respectively), where N indicates the number of components. We compare with a Dirichlet Process Mixture Model (DPMM) [8] as well as the Focused Mixture Model (FMM) [24], a recently proposed Bayesian generative model with state-of-the-art performance. Finally, with compare with OSORT [25], an online sorting algorithm. Only OPASS and OSORT methods were online as we desired to compare to the state-of-the-art *batch* algorithms which use all the data. Note that OPASS algorithms learned $\mathbf{D}$ from the first five seconds of data, whereas all other algorithms used a dictionary learned from the entire data set.

The single-channel experiments were all run on channel 2 (the results were nearly identical for all channels). The spike detections for the offline methods used a threshold of three times the noise standard deviation [5] (unless stated otherwise), and windowed at a size $L = 30$. For multichannel data, we concatenated the $M$ channels for each waveform to obtain a $M \times L$-dimensional vector.

The online algorithms were all run with weakly informative parameters. For the normal-Wishart, we used $\boldsymbol{\mu}_0 = \mathbf{0}$, $\lambda_0 = 0.1$, $\mathbf{W} = 10\mathbf{I}$, and $\nu = 1$ ($\mathbf{I}$ is the identity matrix). The AR process corresponded to a GP with length-scale 30 seconds, and variance 0.1. $\alpha$ was set to 0.1. The parameters were insensitive to minor changes. Running time in unoptimized MATLAB code for 4 minutes of data was 31 seconds for a single channel and 3 minutes for all 4 channels on a 3.2 GHz Intel Core i5 machine with 6 GB of memory (see Supplementary Fig. 11 for details).

**Performance on partial ground truth data** The main empirical result of our contribution is that all variants of OPASS detect more true positives with fewer false positives than any of the other algorithms on the partial ground truth data (see Fig. 1). The only comparable result is the OSORT; however, the OSORT algorithm split the IC cluster into 2 different clusters and we combined the two clusters into one by hand. Our improved sensitivity and specificity is *despite* the fact that OPASS is fully online, whereas all the algorithms (besides OSORT) that we compare to are batch algorithms using all data for all spikes. Note that all the comparison algorithms pre-process the data via thresholding at some constant (which we set to three standard deviations above the mean). To assess the extent to which performance of OPASS is due to *not* thresholding, we implement FAKE-OPASS, which thresholds the data. Indeed, FAKE-OPASS's performance is much like that of the batch algorithms. To get uncertainty estimates, we split the data into ten random two minute segments and repeat this analysis and the results are qualitatively similar.

One possible explanation for the relatively poor performance of the batch algorithms as compared to OPASS is a poor choice of the important—but often overlooked—threshold parameter. The right panel of Fig. 1 shows the receiver operating characteristic (ROC) curve for the k-means algorithms as well as OPASS and MOPASS (where M indicates multichannel, see below for detail). Although we

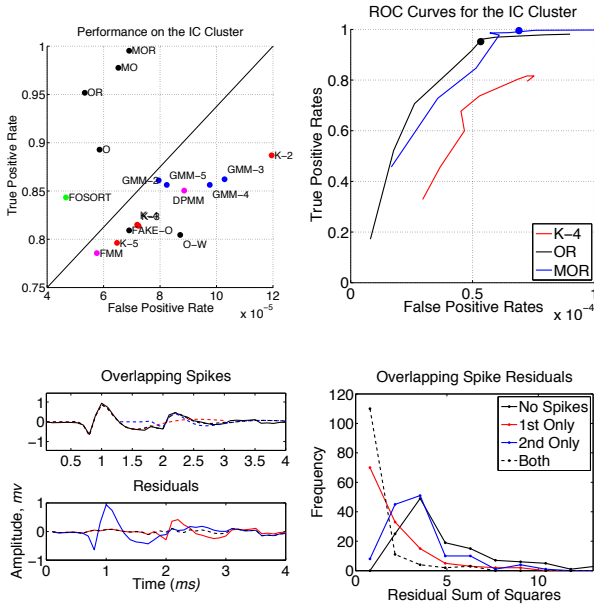

Figure 1: OPASS achieves improved sensitivity and specificity over all competing methods on partial ground truth data. (a) True positive and false positive rates for all variants of OPASS and several competing algorithms. (b) ROC curves demonstrating that OPASS outperforms all competitor algorithms, regardless of threshold (• indicates learning $\Lambda$ from the data).

Figure 2: OPASS detects multiple overlapping waveforms (Top Left) The observed voltage (solid black), MAP waveform 1 (red), MAP waveform 2 (blue), and waveform from the sum (dashed-black). (Bottom Left) Residuals from same example snippet, showing a clear improvement in residuals.

typically run OPASS without tuning parameters, the prior on $\Lambda$ sets the expected number of spikes, which we can vary in a kind of "empirical Bayes" strategy. Indeed, the OPASS curves are fully above the batch curves for all thresholds and priors, suggesting that regardless of which threshold one chooses for pre-processing, OPASS always does better on these data than all the competitor algorithms. Moreover, in OPASSwe are able to infer the parameter $\Lambda$ at a reasonable point, and the inferred $\Lambda$ is shown in the left panel of Fig. 1. and the points along the curve in the right panel. These figures also reveal that using the correlated noise model greatly improves performance.

The above analysis suggests OPASS's ability to detect signals more reliably than thresholding contributes to its success. In the following, we provide evidence suggesting how several of OPASS's key features are fundamental to this improvement.

**Overlapping Spike Detection** A putative reason for the improved sensitivity and specificity of OPASS over other algorithms is its ability to detect overlapping spikes. When spikes overlap, although the result can accurately be modeled as a linear sum in voltage space, the resulting waveform often does not appear in any cluster in PC space (see [1]). However, our online approach can readily find such overlapping spikes. Fig. 2 (top left panel) shows one example of 135 examples where OPASS believed that multiple waveforms were overlapping. Note that even though the waveform peaks are approximately 1 ms from one another, thresholding algorithms do not pick up these spikes, because they look different in PC space.

Indeed, by virtue of estimating the presence of multiple spikes, the residual squared error between the expected voltage and observed voltage shrinks for this snippet (bottom left). The right panel of Fig. 2 shows the density of the residual errors for all putative overlapping spikes. The mass of this density is significantly smaller than the mass of the other scenarios. Of the 135 pairs of overlapping spikes, 37 of those spikes came from the intracellular neuron. Thus, while it seems detecting overlapping spikes helps, it does not fully explain the improvements over the competitor algorithms.

**Time-Varying Waveform Adaptation** As has been demonstrated previously [26], the waveform shape of a neuron may change over time. The mean waveform over time for the intracellular neuron is shown in Fig. 3a. Clearly, the mean waveform is changing over time. Moreover, these changes are reflected in the principal component space (Fig. 3b). We therefore compared means and variances OPASS with OPASSR, which models the mean of the dictionary weights via an auto-regressive process. Fig. 3c shows that the auto-regressive model for the mean dictionary weights yields a time-varying posterior (top), whereas the static prior yields a constant posterior mean with increasing posterior marginal variances (bottom). More precisely, the mean of the posterior standard deviations for the time-varying prior is about half of that for the static prior's posteriors. Indeed, the OPASSR yields 11 more true detections than OPASS.

**Multielectrode Array** OPASS achieved a heightened sensitivity by incorporating *multiple* channels (see MOPASS point in Fig. 1). We further evaluate the impact of multiple channels using a three

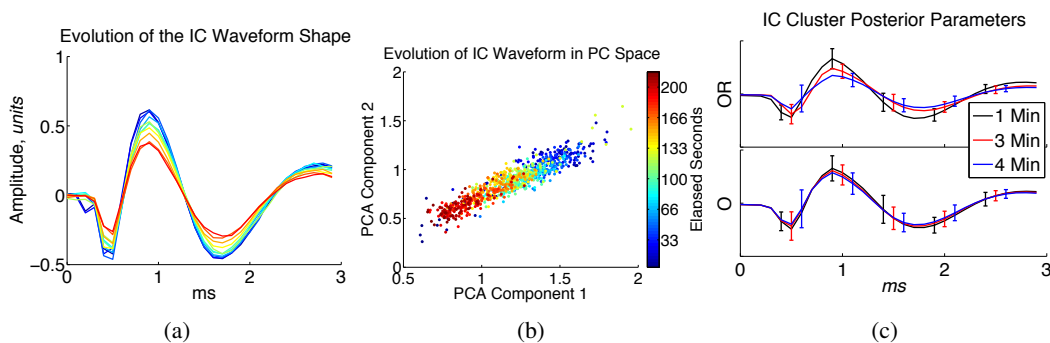

Figure 3: The IC waveform changes over time, which our posterior parameters track. (a) Mean IC waveforms over time. Each colored line represents the mean of the waveform averaged over 24 seconds with color denoting the time interval. This neuron decreases in amplitude over the period of the recording. (b) The same waveforms plotted in PC space still captures the temporal variance. (c) The mean and standard deviation of the waveforms at three time points for the auto-regressive prior on the mean waveform (top) and static prior (bottom). While the auto-regressive prior admits adaptation to the time-varying mean, the posterior of the static prior simply increases its variance.

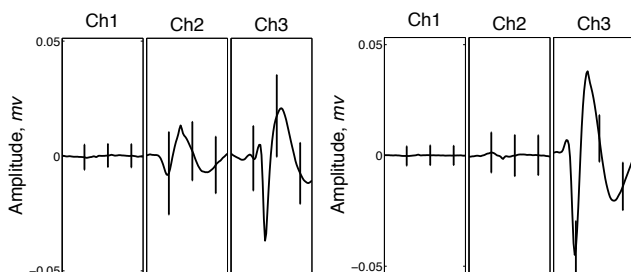

Figure 4: Improving OPASS by incorporating multiple channels. The top 2 most prevalent waveforms from the NeuroNexus dataset with three channels. Note that the left panel has a waveform that appears on both channel 2 and channel 3, whereas the waveform in the right panel only appears in channel 3. If only channel 3 was used, it would be difficult to separate these waveform.

channel NeuroNexus shank (Supp. Fig. 7a). In Fig. 4 we show the top two most prevalent waveforms from these data across the three electrodes. Had only the third electrode been used, these two waveforms would not be distinct (as evidenced by their substantial overlap in PC space upon using only the third channel in Fig. 7b). This suggests that borrowing strength across electrodes improves detection accuracy. Supplementary Fig. 10 shows a similar plot for the eight channel data.

## 5 Discussion

Our improved sensitivity and specificity seem to arise from multiple sources including (i) improved detection, (ii) accounting for correlated noise, (iii) capturing overlapping spikes, (iv) tracking waveform dynamics, and (v) utilizing multiple channels. While others have developed closely related Bayesian models for clustering [8, 27], deconvolution based techniques [1], time-varying waveforms [26], *or* online methods [25, 3], we are the first to our knowledge to incorporate *all* of these.

An interesting implication of our work is that it seems that our errors may be irreconcilable using merely first order methods (that only consider the mean waveform to detect and cluster). Supp. Fig. 8a shows the mean waveform of the true and false positives are essentially identical, suggesting that even in the full 30-dimensional space excluding those waveforms from intracellular cluster would be difficult. Projecting each waveform into the first two PCs is similarly suggestive, as the missed positives do not seem to be in the cluster of the true positives (Supp. Fig. 8b). Thus, in future work, we will explore dynamic and multiscale dictionaries [28], as well as incorporate a more rich history and stimulus dependence.

### Acknowledgments

This research was supported in part by the Defense Advanced Research Projects Agency (DARPA), under the HIST program managed by Dr. Jack Judy.

## Footnotes

[1]Online gamma Process Autoregressive Spike Sorting

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
