[Supplementary Material]

# A Supplementary Text

**Notation:** Unless otherwise specified, lower-case English alphabet characters indicate scalars $x \in \mathbb{R}$. Bold indicates column vectors $\boldsymbol{x} \in \mathbb{R}^p$, and upper-case bold indicates matrices, $\mathbf{X} \in \mathbb{R}^{p \times q}$. Parameters and constants are Greek characters. Time is $t \in [0, T]$, $i \in [N]$ indexes the $N$ neurons, where $[N] = \{1, 2, \ldots, N\}$. Script denotes sets and pipes denote the cardinality of the set, e.g. $|\mathcal{T}|$.

Our overall model is then:

$$\mathcal{T}_i \sim \mathrm{PP}(\lambda_i) \quad i \in \mathbb{N}, \qquad \text{where } \Lambda(\cdot) = \sum_{i=1}^{\infty} \lambda_i \delta_{\theta_i^*} \sim \Gamma\mathrm{P}(\alpha, \mathsf{H}(\cdot|\phi)), \quad (5a)$$

$$x_i(t) = \sum_{j=1}^{|\mathcal{T}_i|} \sum_{k=1}^{K} y_{ijk}^* \mathsf{d}_k(t - \tau_{ij}), \qquad \text{where } \mathbf{y}_{ij}^* \sim \mathsf{N}_K(\boldsymbol{\mu}_i^*, \Sigma_i^*) \quad i, j \in \mathbb{N}, \qquad (5b)$$

$$x(t) = \sum_{i=1}^{\infty} x_i(t) + \varepsilon_t, \qquad \text{where at any time } t, \varepsilon_t \sim \mathsf{N}(0, \Sigma_x) \text{ independently} \qquad (5c)$$

---

**Pseudocode 1** Generative mechanism for the multi-electrode, non-stationary, discrete-time process

---

Input:       a) the number of bins $T$, and the bin-width $\Delta$
         b) the $K$-by-$L$ dictionary $\mathbf{D}$ of $K$ basis functions
         c) the DP hyperparameters $\alpha$ and $\phi$.
         d) the transition matrix $\mathbf{B}$ of the neuron AR process
Output:     An $M$-by-$T$ matrix $\mathbf{X}$ of multielectrode recordings.

1: Initialize the number of clusters $C_1$ to 0.
2: Draw the overall spiking rate $\Lambda \sim \mathrm{Gamma}(\alpha, 1)$.
3: **for** $t$ in $[T]$ **do**
4:      Sample $\widetilde{z}_t \sim \mathrm{Bernoulli}(\Lambda\Delta)$, with $\widetilde{z}_t = 1$ indicating a spike in bin $t$.
5:      **if** $\widetilde{z}_t = 1$ **then**
6:          Sample $\widetilde{\nu}_t$, assigning the spike to a neuron, with $\mathsf{P}(\widetilde{\nu}_t = i) \propto \begin{cases} |\mathcal{T}_i^t| & i \in [C] \\ \alpha & i = C + 1 \end{cases}$
7:          **if** $\nu_t = C_t + 1$ **then**
8:              $C_{t+1} \leftarrow C_t + 1$.
9:              Set $\theta_{C_{t+1}}^* \sim H_\phi(\cdot)$, and $\mathcal{T}_{C_{t+1}} = \{t\}$.
10:         **else**
11:             $\mathcal{T}_{\nu_t} \leftarrow \mathcal{T}_{\nu_t} \cup \{t\}$.
12:         **end if**
13:         Set $\theta_t = \theta_{\nu_t}^*$, recalling that $\theta_t \equiv (\boldsymbol{\mu}_t, \Sigma_t)$.
14:         Sample $\mathbf{y}_t = (\mathbf{y}_t^1; \cdots; \mathbf{y}_1^M) \sim \mathsf{N}(\boldsymbol{\mu}_t, \Sigma_t)$, determining the spike shape at all electrodes.
15:         $x_t^m = \sum_{h=1}^{L} \mathbf{D}_{:,h}^{\mathsf{T}} \mathbf{y}_{t-h-1}^m + \epsilon_t^m$      where $\epsilon_t^m \sim \mathsf{N}(0, \sigma^2), m \in [M]$.
16:         Update the cluster parameters: $\boldsymbol{\mu}_i^* = \mathbf{B}\boldsymbol{\mu}_i^* + r_i \quad i \in [C_{t+1}]$
17:      **end if**
18: **end for**

---

# B  Supplementary Figures

(a)                                    (b)

Figure 5: (a)Dictionary learned from the first 5 seconds of data from the HC1 dataset. (b) Percentage of variance explained by each PCA component.

Figure 6: (a) This shows the average number of true positives versus the average number of false positives in the intracellular cluster for 2 minute segments of the 4 minutes of the experiment. OPASS does better than all the competitors.

Figure 7: Improving OPASS by incorporating *multiple* channels. (a) Three electrode device showing local proximity of electrodes with channel indexes in large, red numbers. (b) The representation of detected spikes on the 3rd channel in PCA space. This cluster does not seem separable here.

Figure 8: False and true positive detections have the same first-order statistics, making detection using only these statistics quite difficult. (a) Errorbar plots of the true positives, false positives, and missed positives in the IC cluster. While the false positives have slightly more variability, the mean shape for the false positives and the true positives is nearly identical. The true misses have a significantly lower amplitude as well as high variability. (b) All waveforms from the IC neuron as well as those we estimated from the IC neuron projected onto the first two PC space.

Figure 9: Pairs plot of true positives (black), false positives (blue ×'s), and missed positives (red +'s). It does not seem like clustering in this space could yield much improvement.

Figure 10: OPASS multielectrode performance. (a) 8 electrode device showing local proximity of electrodes with channel indexes in large, red numbers. (b,c,d) Top three most prevalent waveforms. Each waveform shape is 2 ms long.

Figure 11: OPASS scales linearly with amount of data, with a slope smaller than one, meaning that OPASS can operate in real-time.