[Reviews · NeurIPS 2013]

Submitted by Assigned_Reviewer_1

Spike sorting is an important pre-processing for analyzing electrophysiological data. Authors developed a novel online spike sorting algorithm that has the following features; (1) improved spike detection, (2) accounting for correlated noise, (3) capturing overlapping spikes, (4) tracking waveform dynamics, and (5) utilizing multiple channels. Using two electrophysiological data sets, they showed that the proposed algorithm outperformed previous state-of-the-art methods.

- Quality
Theoretical background of the paper is solid and sound. Their claims are well supported by theoretical analysis and experimental data.

- Clarity
The paper was organized well and is written clearly.

- Originality
The proposed method consists of a novel combination of state-of-the-art statistical methods. The review considers that the paper is highly original.

- Significance
The proposed algorithm has a potential to enable massively parallel recording with adaptive control of specific stimuli. The paper will have a significant impact on the community.

Finally, here is a list of several suggestions for improving the manuscript/future work.

- Comparing the performance by simulated data (such as Quian Quiroga et al., 2004) may further illustrate the strength of the proposed method.

- Line 267:
The length of the 2nd data set is missing.

- Line 277:
To compute D, the authors used the first five seconds of the data. However, the necessary length of data would depend on an activity level of neural signals (firing rate of neurons). It would be interesting if the authors could discuss the relationship.

- Line 284:
IC (intracellular) might be a little confusing because it looks “independent component”. If they could come up a better abbreviation, that would be good.

- Lines 306-209 & Figure 11:
The relationship between algorithm time and data time is not clear. The main text says 31 sec of algorithm time was necessary for processing 4 min of a single channel data. But Figure 11 indicates that 31 sec of algorithm time is necessary for 150 sec of data. Please clarify.

- Lines 356 & 364:
“… one example of “135” examples where OPASS believed that multiple waveforms were overlapping” and “… we detect, “37” of them we believe to be overlapping” are confusing. Please clarify why the numbers are different.

- Line 427:
It is educational if the authors could discuss the number of channels when the slope exceeds 1.

- Line 492:
It should be “Section 2.2”.
Summary: The paper describes a novel online spike sorting algorithm that is capable of learning the number of separable neurons and resolve overlapping spikes. The method is expected to have a major impact to a neuroscience community.

Submitted by Assigned_Reviewer_2

This paper introduces a Bayesian nonparametric method to address the problem of spike detection and sorting (i.e. for jointly detecting neural spikes and assigning them to neurons). These steps are naturally of critical importance to many computational models of neural spiking data as they essentially create the 'ground truth' data on which many models operate. Improving these methods thus is of great interest to the wider computational neuroscience community. The authors develop an efficient online method based on the gamma process that models both the waveforms of spikes for each neuron and the assignment of spikes to neurons. The use of a Chinese restaurant process elegantly allows the model to add new neurons (i.e. the number of neurons from which spikes are detected does not need to be specified a priori). There are some neat features to the model, such as the ability to model non-stationary waveforms (i.e. varying over time). The authors perform an empirical comparison to previous state of the art approaches that provides a compelling case for their method and provide an interesting discussion.

Quality:
The authors derive a model that is conceptually quite complex although neat and appears correct. The novel concepts, methods and empirical analysis are of high quality. The writing should be improved on (this perhaps falls under the clarity category) particularly to clarify the model, although this can be done for a camera ready.

Clarity:
As stated above, the model is rather complex. The authors try to explain the model by starting with simpler models (i.e. a homogenous Poisson process) and then in a step by step fashion add complexity and draw connections to various distributions until they arrive to what their model actually is. I believe this is an attempt to explain their approach to the reader in an accessible way, but I feel it is somewhat confusing. Rather than draw parallels to the entire Bayesian nonparametrics toolbox, it would be better in my opinion to state what the actual model is and then explain the assumptions taken to arrive at this model. I think this is also an opportunity to reclaim a significant amount of space (which is clearly scarce here) which can be used to put in a diagram (graphical model) of the model(s) introduced in this paper.

Originality:
This appears to be original work and addresses some shortcomings of previous approaches. There are similar approaches to spike sorting (e.g. the Dirichlet process mixture models for spike sorting of Gasthaus et al.). However, this paper makes significantly more than delta changes to those works in the model, in how it adapts the waveforms and performs real time spike sorting using variational Bayesian methods.

Significance:
As stated above, spike sorting and detecting are two critical early steps in the computational neuroscience modeling pipeline. Thus improving accuracy would be very useful to many computational models which analyse the data resulting from these steps and improving speed can significantly help achieve real time decoding and move towards larger populations of neurons.

Detailed comments:
48: inference -> infer
52: develop -> developing
58: that that
77: It sounds a bit strange to say that you 'relax' simplifying assumptions - i.e. sounds like you make them simpler
79: as -> are
90: Rather that -> Rather than
163: The mean and covariance are drawn from a Wishart distribution, of which the parameters are drawn from the CRP correct? Perhaps you should add another line in the equation array to make this clear.
183: Twiddles is not a word I am familiar with. Tildes?
184: Associate -> Associated
185: in -> is
197: the subvector
209: I really appreciate Algorithm 1 - I think this should be in the main text if you can find space
224: case a -> case of a
424: being estimates?
428: "embarassingly" parallel sounds rather colloquial and sarcastic

The paper states that code is provided in the supplementary materials to run the models introduced in the paper. I would like to run this code to verify but I can't find it in the supplementaries.

A reference that motivates the importance of waveform non-stationarity.
G. Santhanam, M. D. Linderman, V. Gilja, A. Afshar, S. I. Ryu, T. H. Meng, and K. V. Shenoy. HermesB:
A continuous neural recording system for freely behaving primates. IEEE Transactions on Biomedical
Engineering, 54(11):2037–2050, 2007.

One concern of this reviewer in the nature of the empirical evaluation is that the ground truth is from only on a single neuron from a single data set. Naturally, I sympathize that obtaining such data is rather difficult and we are lucky that even this ground truth data exists and is provided online. However, it does raise a number of concerns - is it possible that we can overfit as a community if we all attempt to achieve state of the art on this single neuron. Furthermore, what is the guarantee that the assumptions made in this work hold for this particular neuron (or a particular class) but don't for others (e.g. we know different classes of neuron exhibit very different patterns of behavior).

It may be useful to state that [23] compares empirically to a number of models that your method doesn't directly compare to such as [27]. So although not comparing directly to that work you are more or less comparing by proxy.
Summary: This paper develops a conceptually complex but appropriate model for joint spike detection and sorting using recent advances in Bayesian nonparametric methods. This is a nice solution to a problem that is of interest to most researchers interested in analyzing neural spike trains from populations of neurons.

Submitted by Assigned_Reviewer_6

The authors describe a new online method for spike sorting called OPASS. They tackle a number of important issues for spike sorting: overlapping spike, changing waveforms over time, real-time spike sorting and compatibility with multiple channels. Such computationally efficient algorithms will be much needed given recent developments in recording technology.
The authors demonstrate convincingly with interesting comparisons the superiority of their method. They combine a bunch of clever tricks and ideas like the color noise modeling through a GP or the changing waveforms. However, the exposition of the method is dense and could be improved (I wouldn’t know where to even start implementing the algorithm!). Given the computational complexity, I am still puzzled how the authors achieved real-time performance. They might want to elaborate on that.
My advice to the authors would be to shorten some of the grandiose statements in the introduction and use the freed space to explain the key ingredients in more detail. As a general remark, I don’t think it is helping NIPS papers to have extensive supplementary material – so the authors might want to shorten theirs. Also, code would be very helpful and should be made available with the paper (I would really love to compare to our algorithms).
Finally, the authors may want to spell check the manuscript.
Summary: Important advance in spike sorting
Author Feedback

Author rebuttal: We thank the reviewers for their positive comments, as well as their suggestions for improvements. Thanks especially for pointing out typos and details of the experiments that weren't clear. Below we've included our responses to the larger concerns/specific questions.

Availability of the code: We haven't made our code public yet because of concerns over the double blind nature of the review but will make it public at the earliest opportunity.

R2: 163: The mean and covariance are drawn from a Wishart distribution, of which the parameters are drawn from the CRP correct?
That is correct.

R2: 209: I really appreciate Algorithm 1 - I think this should be in the main text if you can find space

We agree with this. We will try to shorten the introduction, and include this. We will use this to help clarify inference (we agree with reviewer 3 that this is a bit dense).

R2: It may be useful to state that [23] compares empirically to a number of models that your method doesn't directly compare to such as [27]. So although not comparing directly to that work you are more or less comparing by proxy.

That is a fair point and we will add this to the paper.

R3:- Lines 306-209 & Figure 11:
The relationship between algorithm time and data time is not clear. The main text says 31 sec of algorithm time was necessary for processing 4 min of a single channel data. But Figure 11 indicates that 31 sec of algorithm time is necessary for 150 sec of data. Please clarify.

This axis on Figure 11 was incorrect. It has been remade and will be corrected in the final version. Additionally, we will add the curves for multichannel data.

R3:- Lines 356 & 364:
“… one example of “135” examples where OPASS believed that multiple waveforms were overlapping” and “… we detect, “37” of them we believe to be overlapping” are confusing. Please clarify why the numbers are different.

The algorithm detected 135 total overlapping spikes; for the “intracellular” neuron, 37 of our detections are from overlapping spikes. This means that 37 of the total 135 overlapping spikes corresponded to the “intracellular” neuron.